# In Situ Detection of Complex DNA Damage Using Microscopy: A Rough Road Ahead

**DOI:** 10.3390/cancers12113288

**Published:** 2020-11-06

**Authors:** Zacharenia Nikitaki, Eloise Pariset, Damir Sudar, Sylvain V. Costes, Alexandros G. Georgakilas

**Affiliations:** 1Physics Department, School of Applied Mathematical and Physical Sciences, DNA Damage Laboratory, National Technical University of Athens (NTUA), 15780 Zografou, Athens, Greece; 2Space Biosciences Division, Radiation Biophysics Laboratory, NASA Ames Research Center, Moffett Field, CA 94035, USA; eloise.pariset@nasa.gov (E.P.); sylvain.v.costes@nasa.gov (S.V.C.); 3Universities Space Research Association (USRA), Mountain View, CA 94043, USA; 4Life Sciences Department, Quantitative Imaging Systems LLC, Portland, OR 97209, USA; dsudar@qitissue.com

**Keywords:** colocalization, complex DNA damage, detection of DNA damage, fluorescence microscopy, dSTORM, single-molecule detection, ionizing radiation

## Abstract

**Simple Summary:**

We comparatively discuss all possible methodologies for the complex DNA damage in situ detection. Fluorescent microscopy (FM) is the universal approach utilized in every technique, thus we discuss several FM variants. As image colocalization analysis is involved in the majority of methods, the related coefficients are reported. We envision this work to be a reference for radiobiologists in order to select and orchestrate the appropriate experimental and analysis strategies for optimized DNA damage detection. Last but not least, in the long term to help the incorporation of DNA damage biomarkers in the clinic as useful diagnostic or prognostic indicators.

**Abstract:**

Complexity of DNA damage is considered currently one if not the primary instigator of biological responses and determinant of short and long-term effects in organisms and their offspring. In this review, we focus on the detection of complex (clustered) DNA damage (CDD) induced for example by ionizing radiation (IR) and in some cases by high oxidative stress. We perform a short historical perspective in the field, emphasizing the microscopy-based techniques and methodologies for the detection of CDD at the cellular level. We extend this analysis on the pertaining methodology of surrogate protein markers of CDD (foci) colocalization and provide a unique synthesis of imaging parameters, software, and different types of microscopy used. Last but not least, we critically discuss the main advances and necessary future direction for the better detection of CDD, with important outcomes in biological and clinical setups.

## 1. Introduction

The evolution of detection methodologies of biological material and associated effects of different stressors at the chromosomal or DNA levels has been remarkable over the last two centuries (Figure 1). The primary goal for every organism on earth is to be able to transfer its genetic material, intact without any changes, to the next generation [1]. This maintenance of genomic integrity must be accomplished despite continuous attacks by endogenous and/or exogenous (environmental) agents, i.e., stressors on the cellular DNA, proteins, and lipids [2]. Improving the techniques and methodologies for detecting these various effects at the cellular level is pivotal to the discovery of mechanistic links between the effectors of changes (stressors) and final biological effects, which may be pernicious or positive for the organism. DNA as a biological molecule has been undoubtedly the most frequent target of intensive research since its molecular structure discovery by Watson and Crick [3].

The improvement of detection methodologies has provided a better understanding of the induction and processing of DNA damage, especially complex, with significant impact in the field of cancer research and treatment strategies. Beyond the importance of understanding how DNA damage is converted to unrepaired lesions and mutations or chromosomal instability, several studies have shown an accumulation of oxidative damage or DSBs in a variety of tumor types (reviewed in [4,5,6]). In an earlier in vivo study, Redon et al. underlined the importance of measuring accurately the full pattern of DSB and oxidative non-DSB lesions in tumor-bearing mice [7]. Precision or personalized medicine is considered as the new paradigm in modern medicine and cancer therapy or diagnostics. As discussed in [8], the accurate assessment of DNA damage and repair is crucial for different fields of cancer research, including toxicological assays evaluating the carcinogenic effect of chemical compounds or the efficacy of therapeutic agents.

Detection of endogenous (background) oxidative DNA damage is a complicated analytical problem since damage levels are relatively low, often of the order of one base lesion per 10^6^ normal bases [4] or 1 double strand break (DSB) per whole cell genome (~6 Gbp for a human diploid cell). In addition, since the DNA damage response (DDR) is a dynamic process, measurements have to be performed ideally in situ and in real time, using for example live monitoring.

The first DNA damage detection attempts emerged in the 1960s–early 1970s, when Aaij and Borst used Agarose Gel Electrophoresis (AGE) to separate the different conformations (intact, open circular and linear) of plasmid DNA from several phages [9]. Since then, AGE has been widely used, expanded to mammalian and human DNA, and further improved towards new techniques such as Comet Assay and Polymerase Chain Reaction (PCR). Another primary DNA damage detection technique is the Halo Assay, which is based on the ability of Propidium Iodide property to relax supercoiled DNA. The Halo Assay was firstly introduced by Vinograd et al. (1965) [10] and has progressively evolved to the Fast Halo Assay [11].

Since the major discovery of radiation-induced clustered DNA lesions (within a small region of the DNA like 10–20 bp apart) in 2000 [12], research has progressed intensively in the field of DNA damage detection and analysis with emphasis on clustered DNA damage, which is currently acknowledged as the primary instigator of the detrimental effects of ionizing radiation (IR), such as mutations and genomic instability (for concise reviews please see [13,14,15]). In order to include all the different forms of damage complexity we consider the terms “clustered” and “complex” equivalent. Henceforth, the term complex DNA damage will be used in the text.

The main technique for detecting complex DNA damage has been DNA gel electrophoresis using DNA Repair Enzymes involved for example in base excision repair (BER) pathways, as damage probes. Through their lyase and endonuclease activity, they convert base lesions into single strand breaks (SSB), inducing de novo DSBs and thus DNA fragmentation, which allows the detection of initial base lesions and SSBs through electrophoresis [16]. In plasmid DNA, the creation of de novo SSBs and subsequent DSBs through BER enzyme treatment is a result of the pre-existence of either complex base lesions or base lesions located in opposite strands already carrying a break within a short distance of a few nm (5–10 nm or 10–20 bp) [17]. The use of DNA repair enzymes as damage probes increases significantly the sensitivity of the detection by a factor of 2–5 [18]. DNA denaturation can be induced under alkaline conditions (pH > 13) to convert the double-stranded DNA molecule into single-stranded, or total breaks. However, in this case, DSBs can no more be distinguished from SSBs. One of the main limitations of electrophoresis-based detection of DNA damage is the lack of precision to distinguish clusters of DSBs from single DSBs, which can be overcome by coupling AGE with atomic force microscopy (AFM) to measure the length of each DNA fragment [19].

For many decades, the main tool for estimating the level of DNA damage and estimating exposure doses of radiation has been Cytogenetics [20]. Classic cytogenetics consists of detecting and scoring chromosomal aberrations using optical microscopy. Fluorescence in situ hybridization (FISH) introduced a remarkable advance in Cytogenetics by labeling each chromosome using a fluorescent probe [21]. Although FISH is a powerful tool to detect chromosomal abnormalities, it cannot identify DNA lesions that do not trigger chromosomal rearrangements.

Another strategy is the use of DNA damage sensing proteins that are recruited at sites of DSBs within seconds to minutes following IR exposure, resulting in the formation of microscopically visible nuclear domains referred to as radiation-induced foci (RIF). In mammalian cells, the Rad51 protein was one of the first proteins identified as forming RIF in mitotic and meiotic cells [22,23]. A landmark in the evolution of DNA damage detection was the work of Rogakou et al., which revealed—via electrophoresis—the correlation between the phosphorylation at serine 139 of the histone H2AX with the existence of DSB [24]. Since then, commercially available antibodies targeting the phosphorylated form of histone H2AX have been developed and the γ-H2AX fluorescent foci assay has become the golden standard for DSB level estimation in situ assuming that each focus sits on 1–2 Mbp chromosomal DNA regions [25].

Additional types of RIF have since been characterized. All of them fall into one of these three categories:Foci of proteins recruited to locations of damage sites, such as 53BP1 [26], MRE11, or NBS1 [27,28];Foci of protein that are modified near the damage site, such as phosphorylated H2AX (γH2AX [24]);Foci resulting from both processes, such as phosphorylated ATM (Ataxia Telangiectasia Mutated) [29].

Because foci detection techniques rely on microscopy, their evolution follow advances in fluorescent microscopy technologies. As microscopes and cameras are continuously adapted, and new molecular probes are developed, the performances of DSB detection follow, reaching the single fluorescent molecule limit.

In this review, we critically discuss the major techniques for detection of DNA damage, with an emphasis on associated microscopy methodologies. Figure 1 shows the timeline from the basic observation of chromosomes in plant cells by Karl Wilhelm von Nägeli in 1842 to currently available sophisticated techniques for DNA damage detection.

### 1.1. General Concept for In Situ Detection of Complex DNA Damage

For any method of DNA damage detection, experimental designs have to take into account the following factors, presented in Figure 2:

The source of damage: DNA damage can be induced by irradiation, chemical reagents, or genetic transformations.

The target to be detected: DNA damage detection can be performed by targeting either the damage itself (direct detection) or components involved in the DDR (indirect detection).

The labeling of the target: The target can be labeled either by antibodies conjugated to fluorescent probes for immunofluorescence detection, or by nanoparticles for Transmission Electron Microscopy (TEM) detection. For live-cell imaging, cells can be genetically engineered to produce fluorescent DNA repair proteins.

The imaging technology: For DNA damage detection, the most common imaging technique is fluorescence microscopy (wide field, confocal, or super resolution). Non-photon-based approaches, such as AFM and TEM, have also been successfully demonstrated.

Image analysis: Upon image acquisition, the final step is image analysis. For fixed cells, single images are acquired per field of view (static), while for live cells, each field of view is recorded over time to be able to track the kinetics of the molecules of interest [30]. When studying the synergistic action of multiple DDR components, the use of colocalizing signals can be implemented.

The structure of our review follows Figure 2; we begin with a short introduction to colocalization (§ 1.2) as it represents the main rational for in situ investigation of complex DNA damage. Then we discuss all detection steps from damage induction to image analysis (§ 2–5). Finally, we discuss the implications of implemented DNA damage techniques to support a more global understanding of biological mechanisms of repair.

### 1.2. Colocalization: The Principle behind the In Situ Detection of Complex DNA Damage

Convergent views on the in situ detection of complex damage support colocalization analysis as the only choice. Colocalization relies on the presence of two or more fluorophores at the same location in the cell, which is detected as a spatio-temporal overlap of two or more dyes in a multichannel image. Taking as an example in situ immunofluorescence, Figure 3 illustrates the detection of colocalized signals, arising from the existence of complex DNA lesions, i.e., several types of DNA damage in a small area of DNA or chromosome. When imaging accuracy increases (single molecule detection) colocalization is expressed as the distance between two point signals. Different levels of colocalization define protein interactions (Table 1): the specific cellular organelle in which the protein is recruited at the sub-cellular level, the contact between adjacent proteins at the sub-light microscopic level, and the specific location of inter-protein interaction at the molecular level.

## 2. Damage Induction

The most common source of clustered DNA lesions is ionizing radiation (IR). However, there are many other sources of DSBs, as reviewed by Tang et al. [31], such as endogenous stress, which is typically driven by Reactive Oxygen species (ROS) attack, or programmed recombinational events, which purposely introduce DSBs under specific genomic contexts. In the endogenous type of damage, non-DSB lesions (oxypurine or oxypyrimidine base damage or single strand breaks) are prevalent. In addition, the characteristic difference between IR and any type of endogenous oxidative stress is the lesions’ density [32]. In the case of IR-induced DNA damage, the density, i.e., damage clustering, increases with LET. When studying IR as a source of DNA damage, specific subcellular areas can be targeted using shielding masks and irradiation [33]. Shielding masks such as metal collimators have 1 to 2 μm-wide parallel slots that allow only stripes of the cell nucleus surface to be irradiated [32]. Ion microbeams can focus the irradiation spot size to sub-micrometer levels, generating high local damage density for studying the recruitment of damage repair proteins [34,35]. Laser micro irradiation is also a module integrated in many confocal microscopes, allowing the instant detection of DNA repair, via live cell imaging, by using the same instrument both to irradiate and monitor cellular responses [36,37,38,39].

For the comparison of radiation-induced damage, as well as radiation-triggered DDR mechanisms, other exogenous sources of DNA damage are being used, such as chemical agents and genetic modifications. Chemical exposure is performed by incubation of the sample with genotoxic reagents, such as hydrogen peroxide, or radiomimetic drugs, such as Bleomycin [40], Mitomycin, or Streptonigrin. Genomically integrated constructs induce specific distributions of DSB clusters [41] by introducing sequences recognizable by I-SceI endonuclease [42]. I-SceI is a double stranded DNase with a large and asymmetric recognition site that is not included in human genome. Thus, upon the expression of I-SceI, DSB clusters are induced at specific locations in the genome, by defining the position of I-SceI recognition sites in the transfected plasmid [42,43,44]. The CRISPR/Cas9-based system is another genome engineering tool that can induce DSBs at specific genomic locations [45].

In addition to radiation and chemical exposure, diet and lifestyle factors, in combination with individual demographic parameters and genetic background, are known to significantly influence levels of DNA damage [46]. In particular, increased DNA damage correlates with aging [47], nutritional deficiencies [48], tobacco smoking [49], alcoholism [50], exposure to recreational drugs [51], and intense physical exercise [52,53]. While in the case of radiation exposure, one third of the induced damage is due to direct interaction of radiation with DNA nucleotides [14], diet and lifestyle factors exclusively modulate DNA damage indirectly, through cellular oxidative stress and formation of reactive free radicals, reactive oxygen species (ROS), reactive nitrogen species (RNS), and peroxidized lipids [54].

A common countermeasure to oxidatively-induced DNA damage is the administration of antioxidants that act as free radical scavengers, as well as mitigators of ROS production and lipid peroxidation processes. Antioxidant compounds have the advantage of being widely available in foods and supplements with usually little risk of side effects to patients. In addition to natural antioxidant compounds—for example honey bee products, curcumin, and polyphenols from fruits and vegetables [55]—the FDA-approved amifostine (Ethylol^®^) drug is widely used for its free radical scavenging properties [56]. In addition to the direct elimination of oxidative compounds, another countermeasure approach is to stimulate the cellular expression of natural antioxidants, such as superoxide dismutase (SOD), glutathione peroxidase (GHS), and catalase [57]. Kuefner et al. combined both protective strategies with a formulation of antioxidants and glutathione-elevating enzymes orally administered prior to x-ray irradiation of 25 healthy volunteers [58]. Two different markers of DNA damage (γ-H2AX and 53-BP1 foci) demonstrated the capacity of this radioprotective agent to significantly decrease DNA damage in blood lymphocytes. Thus, accurate and sensitive methods for DNA damage visualization are essential when evaluating countermeasure efficiency.

## 3. Damage Visualization

### 3.1. Entity to Be Detected: Direct (the Damage Itself) or Indirect (via DNA Repair Proteins)

For the purpose of in situ DNA damage detection, the most common way to indirectly visualize the lesion is by labeling one or more DNA repair enzyme(s). The predominant technique is immunofluorescence-based that uses antibodies specific to the target DNA repair enzyme, either labeled by a fluorescence molecule, or detected by a secondary fluorescent antibody. The most common target for immunofluorescence detection of DNA damage clusters is the phosphorylated histone H2AX, i.e., γH2AX (also called γ-H2AX or gamma H2AX). Other top target DNA repair enzymes are RAD51, TP53BP1, phospo-p53, and PARP1 [59].

Another detection strategy is to label the damage itself. In this case, antibodies bind directly onto specific DNA modifications, such as anti- 8-oxodG for the detection of base lesions [60]. Similarly, the TUNEL (TdT-mediated dUTP-biotin nick end labeling) assay detects DSBs by recognizing the 3’-hydroxyl terminal of DNA strains [61].

Finally, Repair Assisted Damage Detection is a fast technique for simultaneous detection of multiple DNA lesions. Its principle is similar to conventional immunofluorescence, but it utilizes enzymes that insert biotinylated deoxyuridine triphosphate [62] at locations of DNA excisions. The restriction of this method is that the different types of DNA damage are not discernible, since only the total cell fluorescence intensity is measured.

### 3.2. Labeling Techniques for Fixed and Live Cells

Based on the cell status, different labeling strategies are applicable. Typically, immunolabeling applies to fixed cells, while transfection-based techniques or fluorigenic dyes can be used for labeling live cells, and monitoring the kinetics of DNA damage induction and repair.

#### 3.2.1. Immunolabeling of Fixed Cells

The term immunolabeling includes a broad spectrum of methods. Here, we summarize some of the most important ones that specifically concern the in situ detection of DNA damage.

##### Fluorophore Conjugated Antibodies for In Situ Immunofluorescence

The predominant method for the in situ complex DNA damage studies is the in situ immunofluorescence. Damaged cells are incubated—prior to fixation—for a specified period of time in order to allow DNA repair to take place. Then, membrane permeabilization takes place, together with incubation with antibody specific for the targets of interest. Complex DNA damage regions are revealed when antibodies are against DNA repair enzymes that belong to different DDR pathways. Antibodies—raised in different species—may be by themselves fluorescent (one step assay) or an extra incubation is needed with a so-called secondary antibody (two step). Secondary antibodies recognize the constant domain of the light chain of primary antibodies as antigen, while their corresponding domain is emitting fluorescence [32,37,63,64,65,66]. Two-step immunofluorescence has the benefit of more than one secondary antibody to be able to bind to the same primary antibody, thus producing enhanced fluorescence signal and facilitating detection process. Appendix A shows the main experimental steps to perform this approach.

##### Nanoparticle Conjugated Antibodies for Transmission Electron Microscopy (TEM) Analysis

Colocalization analysis via TEM follows the same concept like in situ immunofluorescence assay. The notifying difference between the two methods is the labeling molecule; here, antibodies are conjugated with nanoparticles. The exploration of colocalization events is achieved by the use of nanoparticles of dissimilar size [67,68,69,70,71]. Although TEM assay offers great advantage in resolution-down to a scale of a few nm, its protocols are very sophisticated and quite demanding, and it is also not suitable for live cell imaging.

##### Proximity Ligation Assay

Protein colocalization can also be visualized by a single fluorescent dye, unlike traditional colocalization techniques that utilize one dye per targeted enzyme. In the Proximity Ligation Assay (PLA), a fluorescent signal is produced only when the molecules of interest are located in close proximity. In this case, the secondary antibodies, i.e., PLA probes, are not fluorescent by themselves, but they carry single stranded oligonucleotides that can synthesize fluorescent macromolecule in the simultaneous presence of the target enzymes [44,72,73,74]. PLA in principle works when the proteins of interest are located in distances smaller than 40 nm, making it a competitive assay in terms of complex DNA damage detection. Although as Alsermarz et al. report, they might be still some concerns regarding its specificity [75]. In a recent work, the use of PLA between SUMOylated NPM1 and RAP80, and between NPM1 WT and BRAC1 lead to the observations that: (i) NPM1 is a partner protein of hCINAP. (ii) hCINAP regulates NPM1 deSUMOylation during DSB repair. (iii) NPM1 SUMOylation promotes DSB-induced BRCA1 accumulation. (iv) Loss of hCINAP leads to defects in error-free DSB repair HR. (v) Deletion of hCINAP increases NHEJ efficiency. This work suggests that hCINAP is a potential therapeutic target for Acute Myeloid Leukemia given the different response of hCINAP to chemotherapy reagents in normal and AML cells [76]. Their work is an example of the significant outcomes of DNA damage studies for cancer therapeutic strategies.

#### 3.2.2. Live Cell Imaging

Fluorescence labeling of target proteins in live cells enables to visualize the temporal evolution of intracellular distribution and interactions between the different components of the DDR.

##### Encoding Fluorescence Labeled Proteins through Plasmid Transfection

Live cell imaging is based on the fluorescence expression of the components of the DDR. This can be performed by transfecting the cells with adapted plasmid(s) that carry the modified gene(s) of the protein(s) of interest to make them fluorescent, while conserving their initial properties. Such proteins are called recombinant fusion proteins labeled with fluorescent proteins (FP) or auto-fluorescent proteins (AFPs) [77]. Common AFPs are GFP, EGFP, mCherry, RFP, or mNeonGreen. Green Fluorescent Protein (GFP) and GFP-like proteins are most often used for this purpose as fully genetically encoded labels [78]. A nice example of the power of live cell imaging to better characterize DDR across multiple cell generations was done in non-malignant human mammary epithelial cells (MCF10A) expressing histone H2B-GFP and the DNA repair protein 53BP1-mCherry [79]. Using automatic extraction of RIF imaging features and linear programming techniques, this technique enabled to characterize detailed RIF kinetics for 24 h prior to and after exposure to low and high doses of ionizing radiation, showing that DNA repair occurs in a limited number of large repair domains. In an impressive recent work [80], Jacob et al. performed live cell imaging on heavy ion irradiated cells expressing NBS1-GFP or 53BP1-GFP or EGFP-XRCC1. To achieve live cell imaging at early time points post-irradiation (a few seconds), the microscope was placed in the beam line and controlled remotely.

##### Fluorogenic Dyes

In addition to AFPs, which are intrinsically fluorescent proteins, a more recent technology uses extrinsically fluorescent protein tags. Fluorogenic dyes are small monomers that are not fluorescent themselves, but become fluorescent when forming a complex with the molecule of interest. Small membrane permeant antibodies are added to the cell culture at the desired time point of observation to form a fluorescent complex with the target molecule, which is recognized through the tag gene expressed simultaneously as the protein of interest in the genetically-modified target cells [81]. The interaction of fluorogens with their tags can be either covalent or non-covalent. Since non-covalent binding is weaker, it allows the renewal of fluorogen molecules avoiding photobleaching. This technique also provides the fluorescence emission “on demand,” since it allows to switch the signal on and off, by washing out non-covalently associated fluorogens [82], as implemented by several groups for the detection of DNA repair enzymes [83,84,85,86]. On the other hand, covalently interacting tags such as HaloTags, SNAP-tags or CLIP-tags offer higher quantum yield and photostability. These three tags differ in terms of the type of ligand they bind to (fluorogens, dyes, or labels). This availability enables their use in multiplexed (multicolor) experiments.

Fluorigenic dyes (FDs), both covalently and non-covalently interacting, have several advantages compared to auto-fluorescent proteins. They can be used with a broader selection of fluorescent dyes, they provide a strong fluorescence signal since the protein(s) of interest become fluorescent on demand, and they can be used both in live and fixed cells. Note that for DNA damage labeling, FDs have to be cell permeant in order to reach the cell nucleus, while cell impermeant ligands are used to study the cell surface.

##### Encoding Fluorescence Labeled Proteins Using CRISPR/Cas9

The clustered regularly interspaced short palindromic repeats (CRISPR-Cas9)-associated nucleases system is a powerful tool for genetic engineering with a wide spectrum of applications, including cancer immunotherapy [87] and HIV treatment [88], but also live cell imaging [77,89,90,91,92] by labeling genome regions [77,89,90,91]. A recent work of Sharma et al. reveals that almost any intracellular endogenous protein can be fluorescent tagged via CRISP/Cas9 system [92]. The multiple applications of CRISPR/Cas9 system are described in a nature video [93]. Khan et al. [94] introduced endogenously expressed tags for advanced Single Molecule Localization Microscopy (such as halotags), suggesting that the CRISPR mediated labeling (CRISPR-PALM) could have quantitative benefits. In another work, this technology has been used to study the induction and repair of DNA lesions occurring in specified genome sites, by targeting locations of the Cas9 nuclease [95].

##### Fluorescence Recovery after Photobleaching (FRAP)

FRAP enables to monitor the rate of fluorescence recovery over time, within an initially photobleached area to measure the molecular dynamics of the protein of interest, initially fluorescently labeled. Upon photobleaching of the area of interest, fluorescently labeled components originating from non-bleached surroundings are recruited into the bleached area, revealing the protein dynamics. FRAP was used by Asaithamby and Chen to study the molecular dynamics of 53BP1 with similar kinetics observed for both low and high doses of gamma-irradiation [96]. Tobias et al. studied the early responding proteins NBS1 and MDC1 along with 53BP1 in ^48^Ti-irradiated cells, observing that NBS1 recruitment is accelerated by increasing lesion density, which is due to the different binding modes of NBS1, either directly to DSB or to the surrounding chromatin via MDC1 [97]. Aleksandrov et al. managed to study the dynamics of 70 DNA repair proteins using 151 different HeLa Kyoto cell lines. Each cell line was expressing a different EGFP-tagged DNA repair enzyme and micro-irradiated at two nuclear sites in the nucleus before photobleaching one of the two sites [38].

## 4. Imaging: Microscopy and Cameras

As microscopy evolution goes hand in hand with biological discoveries, a brief but comprehensive review is attempted in this paragraph. The reader may also refer to Table 2 for a timeline of basic microscopy discoveries, and associated resolving power, and Appendix A for further details on the following mentioned methodologies. We mainly discuss photon microscopy application, but we also briefly describe photon-based methodologies. The history of photon microscopy began with conventional wide field fluorescence microscopy, followed by confocal microscopy to reach state-of-the-art super resolution microscopy.

### 4.1. Conventional (Widefield) Fluorescence Microscopy and Basic Features

Conventional Microscopy relies on the illumination of the entire specimen simultaneously; thus, it is also referred to as Wide Field Microscopy (WFM), to be differentiated from confocal microscopy, where the specimen is illuminated partially and sequentially. A conventional fluorescence microscope is equipped with a halogen lamp that emits in the visible and UV spectrum and a pair of filters located between the lamp and the objective lens. WFM can be used either in transmission (trans-illumination) or reflection (epi-fluorescence); the second mode being predominant. The excitation filter selects the emission spectrum (~10–30 nm) to illuminate the specimen and excite its fluorophores, while the emission filter selects the wavelengths emitted by the specimen that will enter the camera sensor. WF-transmission M. can be either Bright-field or Dark-field:Bright-field: When the specimen absorbs or scatters some photons and appear darker than its background, which appears bright; in the bright field image, the unscattered (transmitted) photons are selected to the detector and the scattered photons are blocked.In Dark-field mode the unscattered photons are excluded from the aperture and the scattered ones are selected instead. Hence, the areas around the sample do not scatter the light and they will appear dark, while the specimen will appear bright.

In immunofluorescence, the predominant mode of Conventional MS is the Wide-field dark field reflection epi-Fluorescence.

In transmission M light coming from its source is directed to the condenser, where it is gathered and concentrated into a cone and illuminates the specimen. Transmitted light is collected by the objective and it is directed to the detector. On the other hand, in reflection M. the condenser lens cannot be a separate lens, thus, the objective lens takes on a dual role: the objective acts as a condenser for the incident (stimulator) light and it acts as objective for the reflected (emitted) light.

Although WFM is the ancestor of all modalities discussed in the following, its combination with the computational process known as deconvolution can provide impressive details to the resolved image, as demonstrated by Nakajima et al. who were able to distinguish γH2AX core-track from non-track foci in ^56^Fe irradiated human fibroblasts [98]. The basic microscopy properties are illustrated in Figure 4. Among them, the most important is the objective numerical aperture, which defines the resolution, i.e., the minimum distance that the system can resolve. The objective magnification brings the resolved image details to a level of magnification sufficient for human eye detection.

### 4.2. Confocal Microscopy

Contrary to conventional (WF) microscopy, where the entire field of view is illuminated simultaneously, later-developed systems illuminate only a small area at a time, and the final image is gradually generated by post-processing.

In confocal microscopy (CFM), the illumination is performed by lasers, and thus, no excitation filter is needed, since laser produces monochromatic beams, and thus, no filter selection is required. Two pinholes are implemented in confocal microscopes: one between the light source and the specimen, and a second one between the specimen and the detector. The light source pinhole allows only a small spot of the field of view to be illuminated, while the detector pinhole ensures that only the light emitted by the focal plane reaches the detector, which is a key technology to enhance the image quality and overcome the negative contribution of a scattered signal from out of focus plans and neighboring areas in the same field of view, affecting WFM. Laser scanning CFM was the first adaptation of CFM, followed by spinning disk technology, where two or more pinholes are applied to the light source at a time, allowing more than one voxel to be simultaneously illuminated, thus accelerating image acquisition [36]. Conventional WF epifluorescence M, as well as confocal M are “reflection” based. In FM we use “reflection” and “transmission” for simplicity, since in reality it is not the same wavelength that is transmitted or reflected; it is the emitted light that it propagates, instead.

### 4.3. Super Resolution Microscopy (SRM)

Key SRM applications for DNA damage detection can be found here in [99,100,101,102,103,104,105,106].

#### 4.3.1. Super Resolution: Beyond the Diffraction Limit

The term Super Resolution gathers all microscopy techniques able to resolve an image beyond the diffraction limit, which is defined by the theoretical limit of resolution stated by Ernest Abbe on 1873:(1)$$R_{l}=\frac{\lambda }{2ΝA},\;\mathrm{for}\;\mathrm{lateral}\;\mathrm{resolution}\;$$
and
(2)$$R_{axial}=\frac{2\lambda }{NA^{2}}\;,\;\mathrm{for}\;\mathrm{axial}\;\mathrm{resolution}\;$$
where *λ* is the wavelength of the excitation light and *NA* is the Numerical Aperture of the objective. For the current best *NA* of 1.5 and a wavelength in the middle of the visible spectrum (450 nm), this means that R_lateral_ = 150 nm and R_axial_ = 600 nm. These values are the theoretical limits for a perfectly aligned microscopic system with no noise signal from out-of-focus areas, and perfect labeling and mounting. While not practical yet, this limit can be beaten by using metamaterials with negative refracting index, i.e., negative permittivity and magnetic permeability. Lenses made of this type of metamaterials are called super lenses, and they have the characteristic of propagating not only the far-field part of the emitted light, but also the near-field fraction (evanescence waves), which contains information about the object details that are smaller than the diffraction limit. SRM was awarded by the Nobel prize in Chemistry in 2014 [107]. The reader may refer to the review by Schermelleh et al. [108] for detailed information regarding SRM.

#### 4.3.2. Far Field vs. Near Field and the Evanescence Illumination

Far field light is the “ordinary” light which propagates through matter in an unconfined manner. When light strikes the interface between a medium with refracting index n_1_ and a second with a smaller n_2_, at an angle greater than a critical one θ_c_, the phenomenon of total internal reflection takes place. In this case, a small portion of the reflected light energy penetrates through the interface and propagates parallel to the surface in the plane of incidence creating an electromagnetic field in the medium adjacent to the interface. This field is called the evanescent field, and is capable of exciting fluorophores residing in the immediate region near the interface. The characteristic feature of the evanescent field is the fact that its intensity decays exponentially with depth into the medium of propagation:(3)$$I\left(z\right)=I\left(0\right)e^{-z/d}$$
where *d* is the penetration depth (smaller than one wavelength) and *z* is the distance from the interface [109]. The evanescence wave has two advantages: (a) the quick intensity decrease enables to stimulate selectively the fluorophores originating from a thin slice of the specimen and (b) its frequency is much higher than the incident light. These two properties lead to single molecule detection beyond the diffraction limit, used to detect cellular components up to 100 nm from the cellular membrane [104], which is not suitable for DNA damage detection.

#### 4.3.3. Super Resolution: Stochastic Techniques for Single Molecule Detection

While deterministic microscopy approaches relying on the evanescent field are limited to ~200 nm imaging depth, stochastic approaches such as Stochastic Optical Reconstruction Microscopy (STORM) and Photo-Activated Localization Microscopy (PALM) can achieve single molecule resolution based on the stochastic activation of photo switchable fluorophores. They are associated to computational image reconstruction techniques, such as translation microscopy (TRAM) and super-resolution radial fluctuations (SRRF), allowing a resolution power up to seven times higher than any conventional microscope [110].

Direct STORM (dSTORM) was implemented by Varga et al. to reach a resolution of 20 nm and characterize the nucleosome density around γH2AX break sites [111], by Hofmann et al. to visualize the topology of γH2AX foci clusters according to their distance to heterochromatin [112], and by Liddle et al. to visualize γH2AX nanofoci induced by the radiomimetic drug bleomycin and compare their size and density with endogenous γH2AX foci [113].

### 4.4. Non-Photon Microscopy

#### 4.4.1. Scanning Probe

Scanning probe microscopy includes all microscopy techniques that use a physical probe to scan the specimen. SPM is, thus, not limited by diffraction and can reach resolutions down to a few picometers. The most relevant SPM technique for DNA observation is AFM (atomic force microscopy), for which the solution of DNA has to be spread onto an adherent surface prior to scanning, which precludes in situ observations [19,114].

#### 4.4.2. Electron Microscopy: TEM and SEM

##### Transmission Electron Microscopy-TEM

An electron beam transmits through an ultrathin section of the specimen. The interaction between electrons and specimen creates the image contrast. TEM offers an extreme resolution down to the sub-nanometer scale [115]. Despite its significant contribution to bio-research, its applications are limited to fixed or cryogenized samples. In the context of DNA damage research, TEM has been combined with nanoparticle immunolabeling of DNA repair enzymes [67,116].

##### Scanning Electron Microscopy-SEM

SEM uses an electron beam to scan the surface of the sample. Secondary electrons, backscattered electrons, and characteristic X-rays from atoms of the sample contribute to the image creation. Similar to the other scanning probe microscopy techniques, SEM is limited to the study of samples surfaces. Thus, SEM is not suitable for in situ DNA detection, but it can be used in combination with fluorescence microscopy [117]. The reader may refer to the online SCIENCE photo LIBRARY for examples of SEM-based chromosome images [118].

### 4.5. Cameras and Photomultipliers

CCD cameras have been supporting microscopy for decades, with new generation CCD sensors, Electron Multiplying CCD, which offer sensitivity to single photon events [119]. More recently, CMOS circuits have been developed to function either in global shutter mode, where the whole sensor is exposed simultaneously (like for CCD sensors) or in Rolling shutter mode, where sequential image scanning is performed, which protects samples from photobleaching when paired with confocal microscopy. CCD technology is preferred to image ultra-fast phenomena that require simultaneous imaging of the entire image at a specific time point. Both sensors are color blind, i.e., they cannot distinguish the wavelength of the incident photon. This is the reason why in fluorescence microscopy, each wavelength is filtered successively, and pseudocolors are used to produce the final image from superimposed sequences.

Unlike cameras, photomultipliers transform the photons received from the sample into an electrical signal, building the image pixel by pixel, through sequential illumination of small fragments of the optical field when coupled to a point scan system (such as confocal microscopy).

## 5. Image Analysis

The last step for in situ detection of DNA damage is the image analysis. Since the most severe DNA lesion is DSB and their detection is based on the imaging of fluorescent foci (e.g., γH2AX, 53BP1), we begin this paragraph with foci detection techniques, before presenting co-localization approaches, as well as the main associated image analysis tools relative to our scope software.

### 5.1. Cell and Foci Recognition: Criteria, Algorithms, and Transformations

The first step for any image analysis of DNA damage foci is the localization of cellular nuclei, using one of the methods listed in Table 3:

### 5.2. Colocalization Analysis: Coefficients, Parameters, and Methods

As described earlier, fluorescence colocalization is a general biological concept to detect the simultaneous recruitment of multiple types of molecules, labeled with different colors. In this section, some colocalization analysis approaches are presented, divided into two categories: threshold-based and topology-based approaches.

#### 5.2.1. Threshold-Based Colocalization

In the following, we call *A_i_* and *B_i_* the intensity values of voxel i in channels *A* and *B*, respectively, and *a* and *b* the average intensities over the full image in channels A and B, respectively.

i.Pearson Correlation Coefficient:
(4)$$r_{P}=\frac{\Sigma_{i}\left(A_{i}-a\right)\left(B_{i}-b\right)}{\sqrt{\Sigma_{i}{\left(A_{i}-a\right)}^{2}\Sigma_{i}{\left(B_{i}-b\right)}^{2}}}\;$$

This coefficient quantifies the covariance on a pixel by pixel basis. It provides information about the similarity of intensity of each pixel relative to the average signal of the image [124].

ii.(Manders) Overlap Coefficient, *r*:
(5)$$r=\frac{\Sigma_{i}A_{i}\;B_{i}}{\sqrt{\Sigma_{i}{\left(A_{i}\right)}^{2}\;\Sigma_{i}{\left(B_{i}\right)}^{2}}}$$

The values of r range from 0 to 1. It erases irregularities in signal intensities between different components of an image caused by non-uniform labeling, photo-bleaching, or different settings of the amplifiers. This value is strongly influenced by the relative composition of the compared elements of an image [125].

iii.(Manders) Overlap Coefficient, *r*^2^, *k*_1_, and *k*_2_:
(6)$$r^{2}=k_{1}\times k_{2}\;$$ with (7)$$k_{1}=\frac{\Sigma_{i}A_{i}\;B_{i}}{\sqrt{\Sigma_{i}{\left(A_{i}\right)}^{2}}}$$ and (8)$$k_{2}=\frac{\Sigma_{i}A_{i}\;B_{i}}{\sqrt{\Sigma_{i}{\left(B_{i}\right)}^{2}}}$$

Thanks to the use of different coefficients, $k_{1}$ and $k_{2}$, the r coefficient becomes independent of the ratio of the number of objects in compared elements. The coefficients $k_{1}$ and $k_{2}$ are sensitive to differences in the intensity of channel *B* and *A*, respectively [125].

iv.(Manders) *M*_1_ and *M*_2_ overlap Coefficient:
(9)$$r^{2}=M_{1}\times M_{2}\;$$
with
(10)$$M_{1}=\frac{\Sigma_{i}A_{i,coloc}}{\Sigma_{i}A_{i}{}^{2}}$$
(11)$$A_{i,coloc}=\{\begin{array}{c} A_{i}\;if\;B_{i}>0\\ 0\;\;if\;B_{i}=0 \end{array}$$
and
(12)$$M_{2}=\frac{\Sigma_{i}B_{i,coloc}}{\Sigma_{i}B_{i}{}^{2}}\;$$
(13)$$B_{i,coloc}\;=\{\begin{array}{c} B_{i}\;if\;A_{i}>0\\ 0\;\;if\;A_{i}=0 \end{array}$$

$M_{1}\;and\;M_{2}$ are proportional to the amount of fluorescence of the colocalized voxels across both channels of the image, relative to the total fluorescence in the respective channel. These coefficients can be defined even when the signal intensities in the two channels differ significantly [125].

v.Costes’ Automatic Threshold

The Costes’ approach is performed in two successive steps. It first measures the probability (*p*-value) that true colocalization is present in a selected region of the image (*p*-value > 95%). Then, colocalized pixels in the selected region are identified using the two-dimensional histogram of both channels, allowing the computation of the overall fraction of each proteins being colocalized. Thus, each pixel is forced to be classified either as entirely colocalized or entirely non-colocalized signals [126,127].

vi.Van Steensel’s CCF

The Van Steensel’s Cross Correlation Function (CCF) compute the Pearson coefficients while shifting the image from one channel relative to the other one. From the shape of this CCF plot, it can be determined if the signals of the two channels are positively correlated (peak at the center), mutually exclude each other (minimum at the center), or simply overlap randomly (no visible features).

vii.Cytofluorogram

Cytofluorograms were firstly introduced by Demandolx and Davoust [128]. They visualize colocalization by combining the intensity histograms from the two separate channels into one in a two-dimensional scatter plot. Each pixel of the image is placed on the plot based on the grayscale intensity in both colors, which defines the new pixel coordinates. True colocalization is characterized by a cytofluorogram with dots forming a diagonal.

viii.Li’s ICQ

For each image voxel, the Intensity Correlation Quotient (ICQ) checks if the intensities of both channels differ in the same direction from their means. The ratio of the number of voxels that deviate with the same sign in both channels is normalized to take values between −0.5 and +0.5 and indicates how much the signals in the channels A and B colocalize (+0.5), mutually exclude (−0.5), or are randomly distributed (0).

#### 5.2.2. Topology-Based Colocalization

i.R_col_

R_col_, the colocalization ratio, was introduced by Martin et al. [129] as:(14)$$R_{\mathrm{col}}=\frac{2I_{53\mathrm{bp}1}}{{Ι}_{\gamma H2\mathrm{AX}}+I_{53\mathrm{bp}1}}$$
where I_53pb1_ is the average intensity per pixel of the 53BP1 channel for a given γH2AX focus and I_γH2AX_ is the analogous entity in the γH2AX channel. Thus, unlike the previous methods that were defined over the total image, this parameter is defined based on the DSB focus topology. This adaptation enables to remove noise from the portion of the image that is not of interest. Moreover, since the average intensity of the immunofluorescence staining and the acquisition settings may vary between the two channels, the intensities I_γH2AX_ and I_53BP1_ are normalized to the average intensity per pixel across all γ-H2AX and 53BP1 foci in the analyzed image. Based on Equation (13), Rcol = 0 corresponds to no53BP1 signal (I_53BP1_ = 0) in locations of γH2AX foci, and Rcol = 1 corresponds to equal levels of 53BP1 and γ-H2AX expressions within a given γ-H2AX focus. The expression of Rcol can be extended to the detection of any non-DSB repair enzyme in the area of DSB foci. Therefore, Rcol can take the general formula of:(15)$$R_{\mathrm{col}}=\frac{2I_{\mathrm{nonDSB}\;\mathrm{repair}\;\mathrm{enzyme}}}{{Ι}_{\mathrm{DSB}}+I_{\mathrm{nonDSB}\;\mathrm{repair}\;\mathrm{enzyme}}}$$
where I_nonDSB repair enzyme_ and I_DSB_ are the average intensity per pixel for the non-DSB repair enzyme channel and for the DSB repair channel, respectively, in a given DSB focus.

ii.P_clc_

P_clc_, the colocalization parameter, was introduced by our group [130] as:(16)$$P_{clc}=\frac{average\;mean\;Intensity\;of\;nonDSB\;channel\;over\;DSB\;foci\;volume}{average\;mean\;Intensity\;of\;nonDSB\;channel\;over\;the\;rest\;of\;the\;nucleus\;volume}$$
when Pclc tends to zero, reversed colocalization is implied, while when Pclc = ~1, random staining occurs, and for Pclc > 1, true colocalization takes place. This parameter, like Rcol, uses the DSB topology and is very sensitive to colocalization signals, even under high background signal conditions. The need for Pclc resulted from the search for small fluctuations of non DSB repair enzymes with background nuclear endogenous concentrations (APE1, OGG1). To this end, non DSB signal contribution within DSB foci is normalized by the average mean intensity in this channel within the entire nucleus [13,32,130].

iii.Defining an Extra Type of Focus

Another common method to investigate the degree of colocalization is to introduce an additional type of focus called “colocalization focus” that combines the intensity threshold criterion of the second channel and the detection characteristics of a given type of DSB focus. This leads to a degree of colocalization:(17)$$y=\frac{number\;of\;colocalization\;foci\;per\;cell}{number\;of\;DSB\;foci\;per\;cell}$$

### 5.3. Clustering Analysis

Single molecule microscopy reveals the structure of fluorescent DSB foci, as the accumulation of hundreds of nano-foci [104]. Some authors use the term super-foci in order to distinguish conventional microscopy foci from nano-foci [131]. At the nano-foci level, algorithms can distinguish different groups, such as (a) cluster core points (having many neighbors), (b) outliers (reachable from core points but with few neighbors), and (c) noise. An example of such algorithm is DBSCAN (density based spatial clustering of applications with noise) [132], also implemented in ImageJ as part of the BioVoxxel Toolbox (http://www.biovoxxel.de). Super resolution imaging of DSB or complex DNA lesions utilizing DBSCAN algorithm for (γH2AΧ) cluster recognition/cluster classification can be found in [111,112,131].

### 5.4. Co-Localization in Super Resolution Localization Microscopy

The abovementioned co-localization approaches apply to conventional wide field or confocal microscopy, where the smallest γH2AX focus radius is at least 300 nm (restricted from the diffraction limit as well as from Nyquist–Shannon sampling theorem for imaging). However, in SRLM and Single Molecule Microscopy, colocalization is expressed in terms of maximum distance between candidate signal points, which can be defined from the localization precision itself (10–20 nm in [104]) or from considerations of steric hindrance between adjacent molecules (95 nm in [131]).

### 5.5. Software

Here we discuss some open source tools commonly used for the analysis of complex DNA lesions, focusing on colocalization. Software or algorithms for tracking elements in live cell imaging are out of the scope of this review. For further details on this section the reader may refer to the Appendix A and Mascalchi and Cordelières [133].

#### 5.5.1. ImageJ

The dominant player in image analysis is ImageJ, an open source image analysis software, cited over 1 million times. NIH Image, developed in 1987, was the predecessor of ImageJ, while the current version of ImageJ was developed 1997. Since then, many distributions and plugins have been made available [134]. Some of them are specifically dedicated to life science imaging:

JACoP: Just Another Co-localization Plugin is an ImageJ plugin, released in 2006 with different strategies to analyze colocalization [126].

Fiji: Fiji Is Just ImageJ, distributed in 2007, is very popular in life sciences, especially for the analysis of electrophoresis, micronuclei detection, and immunofluorescence [135].

Focinator is an interface in ImageJ that provides selected parameters to allow automated selection of regions of interest (ROIs) defined from the size and circularity of nuclei. The macro offers a combination of multichannel evaluation including colocalization analysis [136].

EzColocalization is a recently developed open source ImageJ plugin [137] that (i) selects individual elements from an image based on physical parameters and signal intensity; (ii) plots the localization patterns of reporters; (iii) measures colocalization of multiple reporters based on the combinations of signal intensity thresholds; and (iv) provides data tables with detailed information on each selected element of the sample.

#### 5.5.2. CellProfiler

CellProfiler^TM^ is a free and open-source software, written in Python and compatible with Windows, Mac, and Linux [138]. It has been cited in more than 7000 scientific papers and is launched more that 120,000 times per year [139]. CellProfiler utilizes Machine Learning and Deep Learning for pattern recognition.

#### 5.5.3. ColocalizR

ColocalizR is a new promising and freely available image-analysis application developed for the high-throughput quantification of co-localized signals on a cell-by-cell basis. Developed in R and compatible with Windows, Linux, and MacOS X, ColocalizR combines pixel-based and object-based approaches, thus improving the assessment of colocalization. ColocalizR automatically segments cells and subcellular compartments, and can also measure morphology and texture features [140].

## 6. Biological and Clinical Importance of Progress in Signal Detection

It is currently more and more accepted that one if not the major determining factor regulating the biological response to IR (BRIR) is the induction of DNA damage, which triggers a variety of responses at the cellular and tissue levels, leading to systemic radiation effects [30,141,142]. The nature of the initial damage is pivotal to determine the BRIR, not only in the area of radiation biology, but also for human pathophysiology induced by high oxidative stress and in general cancer research [143,144,145,146].

As discussed above, understanding the induction of genomic instability, cell death, senescence, and loss of cellular functions requires accurate measurements of the level and type of DNA damage induced by IR or intense oxidative stress, as most biological damage is known to occur from misrepairs and mutational events at the DNA or chromosomal level [147].

Experimental evidence supported by simulation data [148] suggests that even low doses of IR can induce DSBs and other non-DSB lesions in small areas of DNA. Both in vitro and in vivo studies have shown DSB persistence (based on the slow resolution kinetics of γ-H2AX foci), even after doses in the range of computed tomography (CT) scans, well below 100 mGy [149,150]. In addition, low levels of endogenous foci induced by oxidative damage have been reported in a variety of normal [151] and tumor cells [152], while H_2_O_2_ at biologically-relevant concentrations were shown to induce a significant increase in oxidative clustered DNA lesions (OCDLs) [153,154].

Thus, although it is commonly believed that OCDLs are a fingerprint of IR (especially particle irradiation), these results suggest that other synergistic conditions (ROS, oxidative agents, replication stress) can also cause DNA damage converted into DSBs during base excision repair [154]. This genomic instability induced by ‘low’ levels of oxidative stress may be involved in unplanned mutagenesis and the etiology of a wide variety of human diseases like chronic inflammation-related disorders, carcinogenesis, neuro-degeneration, and aging [30].

From a clinical perspective, most of the existing methodologies have focused on individual biomarkers as predictors of radiosensitivity, but none of them have been officially integrated into clinical practice so far [155]. However, it is now commonly accepted that radiosensitivity cannot be predicted using a single biomarker, but requires a combination of biomarkers, which can take into account multiple responses, from mutation induction to molecular processes. To this end, integration with microscopy techniques for colocalization of DSB markers (such as γ-H2AX and 53BP1, or DNA-PKcs, RAD51, BRCA1) is essential to the development of robust biomarkers of radiosensitivity for cancer patients [48,129].

## 7. Conclusions

In this review, we present the available techniques for detecting complex (clustered) DNA damage in cells and tissues, emphasizing the following elements.

### 7.1. Fixed Cells vs. Live Cell Imaging

While immunofluorescence enables fast analysis of large number of cells, live cell imaging is the only solution to achieve specific time resolution and spatiotemporal analysis of molecular dynamics, at the cost of bulky video analysis.

### 7.2. Conventional Wide Field vs. Confocal Microscopy

Confocal microscopy is the preferred method to achieve better resolution, while conventional wide field microscopy offers higher experimental scales through automatized image acquisition protocols.

### 7.3. Super Resolution-Wide Field-Stochastic-Single Molecule for In Situ DNA Damage Detection

Based on our analysis, the best approach for in situ DNA damage detection appears to be dSTORM. dSTROM can be performed both on living and fixed cells and provides the best image resolution when coupled to a robust staining protocol and photo-switch fluorophore cocktail. We should also highlight TEM and PLA as competitive approaches.

### 7.4. The Persistent Weakness of Complex DNA Damage Detection

Despite the constantly improving image resolution offered by novel microscopy techniques, the fundamental limitation of DNA damage detection is defined by the number of DNA repair enzymes that can be simultaneously detected. Given that each DNA repair pathway utilizes dozens of proteins and the fact that in case of complex damage many pathways are simultaneously involved, an exhaustive observation should detect hundreds of different types of molecules (507 in *Homo sapiens* [156]). Up to now, the majority of relevant experiments detect up to three DNA repair enzymes simultaneously, which is a serious technical limitation that needs advancement in the future.

By combining observations of DNA damage induction and repair with non-targeted effects [157], better predictions will emerge for short- and long-term systemic effects of IR based on the knowledge of mechanistic processes.

Based on all the above, we envision that a significant evolution towards multiplex detection would be the utilization of CRISPR technology for halotag encoding. CRISPR offers a great target specificity, which can be combined with halotag’s compatibility with a plethora of fluorogens. In the future this combinational technique might allow simultaneous detection of several DNA repair enzymes.

### 7.5. The Crucial Necessity to Study DNA Damage and Repair

One of the main contributions of DNA damage studies as also discussed in the Introduction, resides in the utilization of DDR knowledge for cancer sensitization in radiotherapy and chemotherapy [158], as recently reviewed in [159]. The accurate knowledge of the damage patterns and levels offers unique opportunities not only towards the advancement of therapy using for example protons or chemotherapy drugs such as bleomycin but also cancer generation (carcinogenesis). The route from the initial damage induction to repair and final outcome such as genomic instability is not always fully understood. All the biological effects of carcinogenic agents on tumor creation can be attributed to the DNA types and levels of damage that they cause and by the errors introduced into the organism’s genome during the cells’ attempted amendment of this damage. Therefore, getting ‘better acquainted’ with the initial cellular DNA damage and overcoming the obstacles of accurate detection will push forward cancer research and therapy including the prediction power (Figure 5).

## Figures and Tables

**Figure 1 cancers-12-03288-f001:**
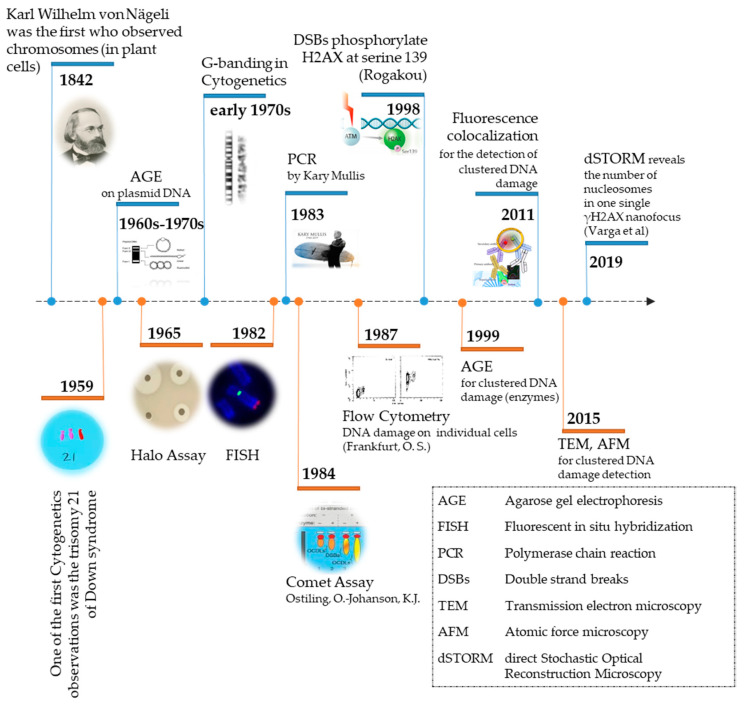
Evolution of experimental methodologies towards the detection of DNA damage at the DNA or chromosomal level from Agarose Gel Electrophoresis (AGE) to Super-Resolution Microscopy.

**Figure 2 cancers-12-03288-f002:**
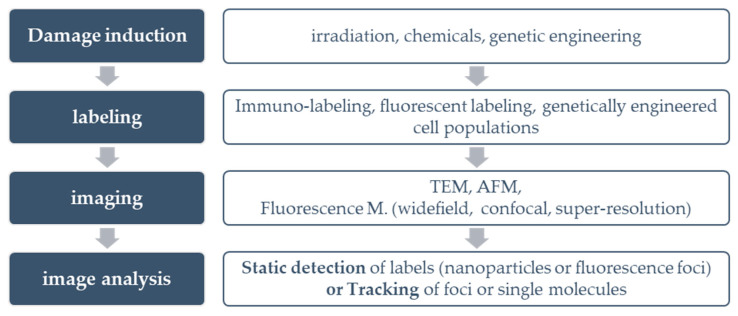
Synopsis of the common fundamental steps across DNA damage in situ detection techniques.

**Figure 3 cancers-12-03288-f003:**
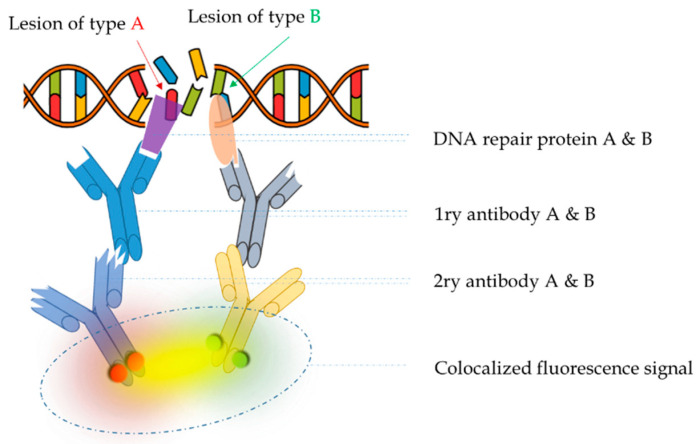
Principle of in situ detection of complex DNA damage via immunofluorescence, assuming the two DNA lesions are of type A and type B, respectively. The cell recruits DNA repair enzymes A and B (purple and salmon shapes) from DNA repair pathway A and B, respectively. Primary antibodies A and B (blue and grey) are specific against enzymes A and B, respectively. Similarly, secondary antibodies A and B (light blue and yellow) are specific against primary antibodies A and B, respectively. Secondary antibodies, upon appropriate stimulation, emit red and green signal, respectively. If the initial lesions A and B are located in close enough proximity, then the green and red fluorescence signals will be colocalized and detected collectively, which will appear as a yellow signal in a multicolor fluorescence image. By reversing this reasoning, the detection of colocalized signals implies the existence of complex DNA damage.

**Figure 4 cancers-12-03288-f004:**
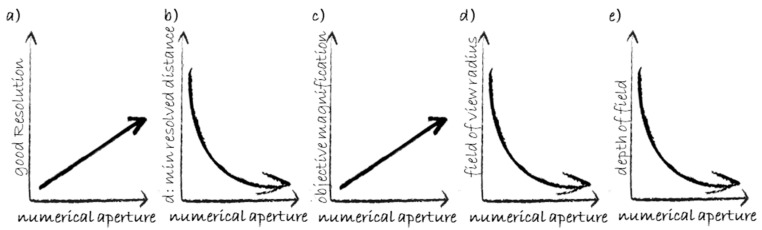
Basic microscopy measures. (**a**) High numerical aperture ensures good resolution (high resolving power). (**b**) The minimum resolved distance “d” between two points of the specimen is the inverse of the resolution. (**c**) Objective magnification usually increases with the numerical aperture. This graph is an empirical approximation. (**d**,**e**) The radius of the field of view as well as the depth of field are inversely proportional to the numerical aperture.

**Figure 5 cancers-12-03288-f005:**
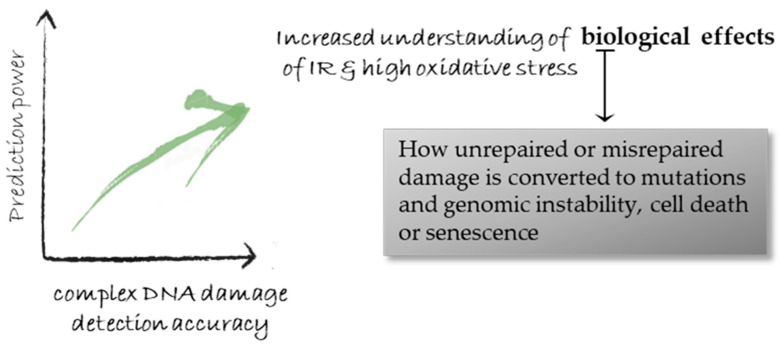
Improved and more accurate detection of complex (clustered) DNA damage leads to a more accurate understanding of the biological effects of IR and cancer (therapy and carcinogenesis), and provides predicting power for models and simulations of radiation-induced risks. The difficulty in improving detection accuracy is considered higher the inherent prediction power limitations.

**Table 1 cancers-12-03288-t001:** Colocalization at different levels of magnitude, and associated detection method.

Level	Question	Detection Method
tissue < 1 mm	Located in the same cell type?	Bright field microscopy
cellular > 10 μm	Located in the same cell?	BFM/Fluorescence microscopy
sub-cellular < 10 μm	Located in the same organelle of a cell?	Fluorescence microscopy/FRET
sub-light microscopic < 200 nm	Existence of a contact between proteins?	FRET/Electron microscopy STED nanoscopy
molecular < 1 nm	Point of contact between proteins?	Electron microscopy/AFM

**Table 2 cancers-12-03288-t002:** Basic microscopy modules, sorted by the year of their first invention. Please note that some of them can be used combinatorically, e.g., PALM/STORM/TRAM/SRRF and LSM (^1^ with tilt series and reconstruction, ^2^ with FIB-SEM).

Yr	Term	Meaning	Current Lateral Resolution	Current Axial Resolution	Wide Field or Point-Scanning	Single Molecule
1931	TEM	Transmission electron M.	50 pm	2 nm ^1^	Point	Y
1937	SEM	Scanning electron M.	0.4 nm	2 nm ^2^	Point	N
1957	CM	Confocal M.	250 nm	600 nm	Point	Ν
1967	LSC	Laser scanning confocal M.	240 nm	600 nm	Point	Ν
1976	FRAP	Fluorescence recovery after photobleaching	250 nm	1 µm	Both	N
1981	SPM	Scanning probe M.	2 pm	0.2 pm	Point	Y
1982	STM	Scanning tunneling M.	0.1 nm	10 pm	Point	Y
1984	SNOM	Scanning near field optical M.	<50 nm	2 nm	Point	N
1985	AFM	Atomic Force M.	<1 nm	0.2 pm	Point	Y
1990	2PEF	2 Photon excitation fluorescence M.	300 nm	500 nm	Point	N
1990	PSTM	Photon scanning tunneling M.	10 pm	0.3 pm	Point	Y
1993	FCS	Fluorescence correlation spectroscopy	1 µm	1 µm	Point	N
2006	STORM	Stochastic optical reconstruction M.	<30 nm	10 nm	Wide	Y
2006	PALM	Photoactivated localization M.	10 nm	10 nm	Wide	Y
2008	3D SIM	3D Structured illumination M.	100 nm	250 nm	Wide	N
2009	SPIM or LSM (LSFM)	Selective plane illumination M. Light sheet (fl.) M	300 nm	800 nm	Wide	N
2010	Bessel LSM	Bessel light sheet M.	300 nm	800 nm	Wide	N
2014	Lattice LSM	Lattice light sheet M.	75 nm	100 nm	Wide	N
2015	STED nanoscopy	Stimulated Emission Depletion	80 nm	100 nm	Point	Y
2016	TRAM	Translation M.	7-fold res improv.	7-fold	Wide	N
2016	SRRF	Super-resolution Radial Fluctuations	Depends on microscopy		Both	Y

**Table 3 cancers-12-03288-t003:** Nuclei detection: criteria, algorithms, and transformations.

#	Name	Objective	Concept	Ref.
1	Local Maximum (algorithm)	Find the pixel(s) with maximum intensity in a delimited area.	Intensity comparison of neighboring pixels to select the pixel(s) with the higher intensity value	
2	Intensity threshold (criterion)	Distinguish the signal from the background to define the boundary of a (convex) object.	Intensity comparison of neighboring pixels; which edge pixels should be discarded as noise and which should be retained.	
3	Area growing (algorithm)	*a* Define the boundary of a (convex) object. *b* Distinguish signal from the background.	*a* The area is defined from a pixel of local maximum intensity (seed) and continuing including neighboring pixels until reaching a specified intensity threshold value. *b* Same process beginning from the local minimum intensity pixel(s).	[120]
4	(white) Top hat (transformation)	Extract small elements or details of an image.	Selects objects that are smaller than a “structuring element” ^1^ and brighter than their surroundings	
5	H-dome (transformation)	Extract small elements or details of an image.	Excludes background by keeping local maxima above an intensity threshold defined from the local background, rescaling the intensity values by subtracting local background levels of each “dome.”	[121]
6	A Trous Wavelet	Enhance contrast in the image by reducing noise from non-specific signals using pattern recognition algorithm.	*a* Creates an intensity-independent image. *b* Applies a constant threshold on the wavelet filtered image. *c* Watershed algorithms separate adjacent spots, based on the aminimum size threshold.	[122,123]

^1^ Structuring element: the size of s.e. is the variable of the transform.

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
