# Peer review of "In Situ Detection of Complex DNA Damage Using Microscopy: A Rough Road Ahead"

_cancers, 2020, doi:10.3390/cancers12113288_

Round 1

Reviewer 1 Report

The review “In situ detection of complex DNA damage using microscopy: A rough road ahead” by Georgakilas and colleagues present a comprehensive analysis of the strengths and weaknesses of the methodologies used to detect DNA damage. The brief history provides necessary context and is useful to orient the readers. There are a few suggestions below that could be consider to strengthen the manuscript

  • Figure 3 is an important concept which seems to get lost in the complexity of the presentation. A more simplistic figure might get the point across better.  It is hard to determine that in fact the nuts are on a DNA.    
  • The proximity ligation assay is presented as a method for detection. While this is possible, the real utility of PLA lies in the ability to detect co-localization of 2 or 3 molecule entities.  Expanding this section to highlight this point would be worthwhile.  
  • The discussion of fluorescent tagging of proteins could be extended to include halotags. This methodology offers some advantages to traditional GFP tagging and is also compatible with Crispr/Cas9 editing systems.

Author Response

Comments and Suggestions for Authors

The review “In situ detection of complex DNA damage using microscopy: A rough road ahead” by Georgakilas and colleagues present a comprehensive analysis of the strengths and weaknesses of the methodologies used to detect DNA damage. The brief history provides necessary context and is useful to orient the readers. There are a few suggestions below that could be consider to strengthen the manuscript

  • Figure 3 is an important concept which seems to get lost in the complexity of the presentation. A more simplistic figure might get the point across better.  It is hard to determine that in fact the nuts are on a DNA.    

Response: We modified Figure 3 for simplification and clarity, and adjusted the caption accordingly.

  • The proximity ligation assay is presented as a method for detection. While this is possible, the real utility of PLA lies in the ability to detect co-localization of 2 or 3 molecule entities.  Expanding this section to highlight this point would be worthwhile.  

Response: We agree with the reviewer, that this is the concept of PLA. We have expanded the corresponding section with an example of using PLA to study DNA repair with potential application to AML therapy. (Xu et al, 2020)

  • The discussion of fluorescent tagging of proteins could be extended to include halotags. This methodology offers some advantages to traditional GFP tagging and is also compatible with Crispr/Cas9 editing systems.

Response: The paragraph withFluorogenic dyes” heading was moved up, in order to follow “Encoding fluorescence labeled proteins through plasmid transfection” paragraph. We agree that halotags, along with SNAP-tag and CLIP-tag that are covalently interacting tags, provide several advantages, which we added to the manuscript. Moreover, CRISPR/Cas9 section was expanded. We thank the reviewer for this important comment.

Reviewer 2 Report

This review reports in situ detection of complex DNA damage using microscopy, especially, technical points of the detection, visualization, and analysis of the complex DNA damage. These information are useful to detect and/or analyze complex DNA damage. This article is acceptable for publication after minor revision. The minor comments are listed below.

  1. The relation between cancer and complex DNA damage should be stated in Introduction.
  2. The number 6.5 Cameras and photomultipliers should be replaced as number 4.5. (line 447)
  3. Figure 5 should be removed because this figure is confusable as the prediction power having linear relation to the complex damage. However, the contents in the figure are important for understanding the biological effects of IR. These matters would be stated in the manuscript.

Author Response

Comments and Suggestions for Authors

This review reports in situ detection of complex DNA damage using microscopy, especially, technical points of the detection, visualization, and analysis of the complex DNA damage. These information are useful to detect and/or analyze complex DNA damage. This article is acceptable for publication after minor revision. The minor comments are listed below.

  1. The relation between cancer and complex DNA damage should be stated in Introduction.

Response: We have added a paragraph in the Introduction and Conclusions,  new paragraphs to meet reviewer’s suggestion.

At this point please allow us to emphasize that we believe very much in the importance of complex DNA damage and its implications to cancer (generation and treatment) but at one hand, we have discussed this recently in Mavragani et al. (2019) Ionizing Radiation and Complex DNA Damage: From Prediction to Detection Challenges and Biological Significance. Cancers, 11, 1789. And earlier in previous studies and review articles from group. Secondly, there is rather limited absolutely straightforward data on the direction connection of complex DNA damage and cancer incidence. Rather more data exists on the strong association between complex damage and mutation or genomic instability.

These thoughts we include in our response also in the manuscript.

  1. The number 6.5 Cameras and photomultipliers should be replaced as number 4.5. (line 447)

Response: Done, thank you.

  1. Figure 5 should be removed because this figure is confusable as the prediction power having linear relation to the complex damage. However, the contents in the figure are important for understanding the biological effects of IR. These matters would be stated in the manuscript.

Response: We thank the reviewer for this comment. We would like to keep this figure underlying the importance of proper and optimized detection of DNA damage. But in any case, we have revised it. We have replaced the linear vector with a “hand-drawn” one, which resembles the form of “y=sqrt(x)” . It is hand-drawn to imply its empirical / approximate nature. The used vector corresponds to increasing monotonicity, while we have chosen to have the arrow like this to emphasize that the difficulty in improving detection accuracy is considered higher the inherent prediction power limitations.

Submission Date

09 October 2020

Date of this review

20 Oct 2020 22:36:48

Reviewer 3 Report

The review article is well written and describes a lot of physical concepts in DNA damage. However it lacks fully connects in DNA damage and stated in the title. the overall manuscript would improve by including more general information about the concepts of DNA damage. 

Especially for a cancer journal it is missing some examples of techniques and their achievements in the past. For example which important discovery in the cancer field each technique is famous for as well as the benefits as well as disadvantages. 

Minor comment: - line 285-6 "In the future this technique might be able to also encode fluorescently tagged DNA repair enzymes." I would exclude from the main test and maybe add it as an outlook in the conclusion paragraph.  

Author Response

Comments and Suggestions for Authors

The review article is well written and describes a lot of physical concepts in DNA damage. However it lacks fully connects in DNA damage and stated in the title. The overall manuscript would improve by including more general information about the concepts of DNA damage.

Response: We agree and we have addressed this comment in the section “2. Damage induction” within of course two limitations: i. the central emphasis of this current work on the complex DNA damage detection and 2. Our previous engagements in research or review manuscripts like Mavragani, I., et al. (2017) Complex DNA Damage: A Route to Radiation-Induced Genomic Instability and Carcinogenesis. Cancers, 9, 91. And also Nikitaki, Z. et al. (2015) Stress-induced DNA damage biomarkers: applications and limitations. Frontiers in chemistry, 3, 35.

Especially for a cancer journal it is missing some examples of techniques and their achievements in the past. For example which important discovery in the cancer field each technique is famous for as well as the benefits as well as disadvantages.

  • Response: We thank the reviewer for this comment. We discussed these applications in a new introductory paragraph, in the last section of the paper “Biological and Clinical importance of progress in signal detection”, and in the conclusion.

Minor comment: - line 285-6 "In the future this technique might be able to also encode fluorescently tagged DNA repair enzymes." I would exclude from the main test and maybe add it as an outlook in the conclusion paragraph.

Response: We have addressed this comment by adding a small paragraph at the Conclusions section.   

Round 2

Reviewer 3 Report

I thank the authors for including all my comments and suggestions. I have no further comments.